# Cardiovascular Disease Burden Attributable to High Sodium Intake in China: A Longitudinal Study from 1990 to 2019

**DOI:** 10.3390/nu16091307

**Published:** 2024-04-26

**Authors:** Liying Jiang, Wanying Shen, Anqi Wang, Haiqin Fang, Qihe Wang, Huzhong Li, Sana Liu, Yi Shen, Aidong Liu

**Affiliations:** 1Jiading Central Hospital, Shanghai University of Medicine & Health Sciences, Shanghai 201899, China; j_meili@126.com; 2Department of Prevention Medicine, College of Public Health, Shanghai University of Medicine & Health Sciences, Shanghai 201318, China; 3Department of Epidemiology, School of Public Health, Nantong University, Nantong 226019, China; shenwanying-@outlook.com; 4Graduate School, Shanghai University of Traditional & Chinese Medicine, Shanghai 201203, China; wanganqi202309@163.com; 5Department of Nutrition Division I, China National Center for Food Safety Risk Assessment, Beijing 100022, China; fanghaiqin@cfsa.net.cn (H.F.); wangqihe@cfsa.net.cn (Q.W.); lihuzhong@cfsa.net.cn (H.L.); liusana@cfsa.net.cn (S.L.); 6National Institute for Nutrition and Health, Chinese Center for Disease Control and Prevention, Beijing 100050, China

**Keywords:** high sodium intake, cardiovascular disease, burden of disease, China

## Abstract

Background: Overconsumption of sodium has been identified as a key driving factor for diet-related cardiovascular diseases (CVDs). China, being a country bearing a hefty burden of CVD, has a large population with diverse cultural traditions and ethnic beliefs, which complicates the patterns of dietary sodium intake, necessitating a systematic investigation into the profile of the high sodium intake (HSI)-related burden of CVD within its subregions. This study aims to estimate the evolving patterns of HSI-induced CVD burden across China from 1990 to 2019. Methods: The methodology used in the Global Burden of Disease Study was followed to assess deaths and disability-adjusted life years (DALYs) by age, sex, region, and socio-demographic index (SDI). The estimated annual percentage change (EAPC) was calculated to quantify the secular changes in the age-standardized mortality rate (ASMR) and age-standardized DALY rate (ASDR). Results: In 2019, 0.79 million deaths and 1.93 million DALYs of CVD were attributed to HSI, an increase of 53.91% and 39.39% since 1990, respectively. Nevertheless, a downward trend in ASMR (EAPC: −1.45, 95% CI: −1.55, −1.35) and ASDR (EAPC: −1.61, 95% CI: −1.68, −1.53) was detected over time. ASMR and ASDR were higher for males, individuals aged ≥60 years, and regions with low-middle SDI. A markedly negative association between the EAPC in both ASMR and ASDR and the SDI was found in 2019 (ρ = −0.659, *p* < 0.001 and ρ = −0.558, *p* < 0.001, respectively). Conclusions: The HSI-induced CVD burden is gender-, age-, and socioeconomic-dependent. Integrated and targeted strategies for CVD prevention are anticipated in the future throughout China.

## 1. Introduction

Cardiovascular diseases (CVDs), representing a group of circulatory system disorders that mainly include heart and vascular diseases, have been the leading cause of morbidity and mortality worldwide over the past decades, and this trend has carried over to now. The newly updated Global Burden of Disease (GBD) Study 2019 assessed that an estimated 18.6 million deaths and 393 million disability-adjusted life years (DALYs) resulted from CVD, accounting for 33% and 15.5% of total deaths and DALYs around the world, respectively [1]. Despite all the endeavors to bring the growing burden of CVD into greater relief, the future trend of it seems still not optimistic. Currently, the pace of decline in CVD prevalence has already decelerated in high-income countries, while the upward trend still persists in low- and middle-income countries [2]. China bears the world’s heaviest burden of CVD [3]. The estimated number of deaths resulting from the disease doubled from 2.42 million in 1990 to 4.58 million in 2019, chalking up rapid and substantial growth in this country [4]. In 2019, CVD was responsible for 46.74% and 44.26% of deaths in rural and urban regions of China, being the cause of death in 2 out of every 5 cases [5]. In the face of population growth and aging, China’s burden of CVD is forecast to rise in the years ahead. A study indicated that, taking only the demographic shifts into account, the annual CVD incidents in China are projected to rise by over 50% between 2010 and 2030, and the current trend of risk factors will further the burden during this period [6].

Primary prevention, focusing on modifiable risk factors, was considered a promising way to reduce the incidence of CVD and promote its subsequent prognosis. Risk factors confirmed to be related to CVD include diabetes, hypertension, dyslipidemia, obesity, smoking, a lack of physical activity, and an unhealthy diet [7]. Unhealthy diets are one of the most relevant modifiable risk factors for CVD. High sodium intake (HSI), as an unhealthy dietary practice, has shown a consistent association with this disease in previous studies [8,9,10]. Limiting daily sodium intake can effectively lower the risk of elevated blood pressure and subsequent CVD, while HSI may have the opposite effect through pathways including stiffened endothelial cells, thickened resistance arteries, and blocked nitric oxide synthesis [11,12]. The risk of fatal and non-fatal CVD is reckoned at 17% for every 1000 mg/d increase in sodium excretion, according to a cohort analysis [13]. Patients with established CVD may be particularly vulnerable to excess sodium, as the risk seems to be greater for those with prior cardiovascular events, heart failure, and diabetes [14]. Therefore, it is important for people, especially those patients, to consume a safe amount of sodium. The World Health Organization (WHO) advises adults to limit their daily sodium intake to below 2000 mg [15]. The American Heart Association proposes that the general public should aim for a sodium consumption of less than 2400 mg/d, whereas individuals with or at a high risk of CVD should aim for less than 1500 mg/d [16].

Although interventions such as legislative policies and awareness campaigns aimed at reducing sodium intake have been widely implemented, current sodium consumption remains beyond ideal, as evidenced by the estimated global average sodium intake of 4310 mg/d.15 Worse still, according to the China National Nutrition and Health Survey (CNNHS), the average daily sodium intake of Chinese adults was 5013 mg/d, far exceeding the recommended level [17]. The sodium intake among Chinese residents varies greatly on temporal and spatial scales, and the dynamic changes in the CVD burden attributable to HSI in China remain to be elucidated. On this basis, this study evaluated the spatiotemporal trends in HSI-related CVD burden in China from 1990 to 2019. Recognizing these epidemiological characteristics and identifying the potential driving factors could lead to a better understanding of the disease burden resulting from HSI in China and serve to formulate evidence-based solutions and control policies tailored to regional-specific needs, thus responding to the call to reduce the CVD burden globally.

## 2. Materials and Methods

### 2.1. Study Data

The Global Burden of Diseases, Injuries, and Risk Factors Study provided a systematic assessment of health loss for 369 diseases and injuries, age-, and sex-specific mortality for 286 causes, and 87 risk factors in 204 countries and territories from 1990 to 2019, with a wide array of standardized analytical procedures, including data screenings, data adjustments, and DisMod-MR (a Bayesian meta-regression tool) estimations to improve the quality and comparability of the data [1]. Through the Global Health Data Exchange Tool (https://vizhub.healthdata.org/gbd-results/ (accessed on 1 August 2023)), we obtained data on deaths, DALYs, age-standardized mortality rates (ASMR), and age-standardized DALY rates (ASDR) of CVD attributable to HSI from 1990 to 2019, which were processed primarily based on raw data from original sources, including the Disease Surveillance Points system in China, the cause-of-death reporting system collected by the Chinese Center for Disease Control and Prevention, medical certification of causes of death for Macao and Hong Kong, and nationwide surveys [18]. Details regarding sodium intake and the incidence and mortality of CVD across China, spanning from January 1990 to December 2019, were obtained from a database compiled by the China National Center for Food Safety Risk Assessment (NFSA) from January 1990 to December 2019 to quantify the estimates of HSI-related CVD burden by age, sex, year, and region and assess its relationship with the sociodemographic index (SDI). This analysis was conducted on 33 administrative units at the provincial level, encompassing 31 provinces, municipalities, and autonomous regions, as well as the Special Administrative Regions (SAR) of Hong Kong and Macao.

As a compound indicator, SDI is measured by the equally weighted geometric mean of lag-distributed income per capita, average years of education for those aged 15 and above, and total fertility rate among women below the age of 25, thus reflecting the socioeconomic level relevant to the health of a certain geographic area [1,19]. It ranges from 0 to 1, representing the varying level of health development from minimum to maximum. The reference SDI quartile as well as the SDI values estimated for China were retrieved in GBD 2017. The national SDI in China was 0.71 in 2017 and its provincial SDI, ranging from 0.47 to 0.86, can be categorized into 4 levels: high SDI region (Beijing, Shanghai, Hong Kong, and Macao), high-middle SDI region (Tianjin, Hebei, Fujian, Guangdong, Jilin, Liaoning, Shandong, Jiangsu, Zhejiang, Heilongjiang, and Inner Mongolia), low-middle SDI region (Tibet, Gansu, and Guizhou), and middle SDI region (all remaining regions) [20].

Since the database is publicly accessible and does not include identifiable information, there is no need for approval by the Research Ethics Committee.

### 2.2. Definitions of Dietary Risk Factors and Associated Outcomes

Sodium intake assessment among the Chinese population in GBD 2019 has been described previously [19,21]. Briefly, DisMod-MR was utilized to convert sodium intake acquired from 24 h dietary recalls and food frequency questionnaires in nine representative Chinese studies, including the China Nutrition Survey in 1992 and 2002 and the KaiLuan Study during 2006–2016, into 24 h urinary sodium to reduce the bias (the gold standard data source for GBD 2019). A spatiotemporal Gaussian process regression framework was employed to estimate the mean sodium intake by age, sex, year, and region. HSI was defined as sodium intake (in grams per day) greater than 3 g/d (95% UI: 1 g, 5 g). The International Statistical Classification of Diseases and Related Health Issues-10 (ICD-10) was used by GBD to identify the cases of CVDs, and codes for 11 CVD subcategories were presented in Appendix A.

### 2.3. Estimation of High Sodium Intake-Attributed CVD Burden

Methods employed in GBD 2019 to estimate the risk-attributable disease burden have been expounded upon elsewhere [1,19]. In brief, the proportion of preventable disease burden was quantified for each risk factor had the level of exposure been maintained at the level linked to minimum risk, which was defined as the theoretical minimum-risk exposure level (TMREL). Founded on exposure data of each risk factor released in the large-scale population surveys and reports, DisMod-MR and spatiotemporal Gaussian process regression were utilized to synthesize data and determine the exposure level of risk and its TMREL. The relative risk (RR) for the HSI-CVD pair was summarized as a function of exposure based on the meta-analysis of prospective observational studies published globally. Population-attributable fractions (PAFs), presenting the proportion of CVD burden that might be reduced within a given timeframe and population if the exposure to HSI in the past was lowered to the counterfactual level of the TMREL, were calculated for the HSI-CVD pair by age, sex, and year by using exposure level estimates of RR and TMREL. The standard GBD PAF formula is outlined as follows:$${PAF}_{asgt}=\frac{\sum_{x=l}^{u}RR_{asg}\left(x\right)P_{asgt}\left(x\right)-RR_{asg}\left({TMREL}_{as}\right)}{\sum_{x=l}^{u}RR_{as}\left(x\right)P_{asgt}\left(x\right)}$$
where *PAF_asgt_* represents the PAF for CVD burden attributable to HSI for age group a, sex s, geographic area g, and year t. *RR_ast_* (x) denotes the *RR* between exposure level x (from l to u) of HSI and CVD for age group a, sex s, and year t; and *P_asgt_* (x) stands for the proportion of the population exposed to HSI at the level x for age group a, sex s, geographic area g, and year t. *TMREL_as_* refers to the TMREL for age group a, and sex s.

DALYs were calculated with a sum of years of life lost (YLLs, multiplying observed deaths from each cause in each age group by the age-specific standard life expectancy estimated using life table methods) and years lived with disability (YLDs, years lived with any short-term or long-term health loss weighted for severity by the disability weights). Then, the burden of CVD attributable to HSI was estimated by multiplying the corresponding PAF for each age group, sex, year, and geographic area with the total number of CVD deaths and DALYs.

### 2.4. Statistical Analyses

To remove the effects of the differences in age structure within a population, age-standardized rates (ASRs) would be necessary. Hence, data on age-standardized mortality and DALY rates were reported in numerical form with 95% uncertainty intervals (UIs) based on the interquartile range of the 25th and 75th ranked estimates across 1000 samples. The PAF was calculated to measure the contribution of HSI exposure to the subsequent occurrence of CVD-related burden using a comparative risk assessment framework. The estimated annual percentage change (EAPC), which indicates the trajectory of ASRs over a specified interval, was applied to reflect the secular trends in ASMR and ASDR of CVD between 1990 and 2019.

A linear regression model was fitted to the logarithmic age-standardized indicators: ln(ASR) = α + βx + ε, where x was the calendar year. After that, EAPC and its 95% confidence interval (CI) could be derived on a log scale from the model 100 × [exp(β) − 1]. It is deemed indicative of a rising trend in ASRs of deaths and DALYs if the lower limit of the 95% CI for the corresponding EAPC estimation was higher than 0, and the reverse is true when the upper limit of the 95% CI was lower than 0. Otherwise, the ASRs were considered to be stable.

To investigate the association between the CVD burden attributable to HSI and the socioeconomic development across various provinces, relationships between the ASRs of deaths and DALYs and the SDI from 1990 to 2019 were examined by smoothing spline models. Spearman’s test was used to explore the potential factors influencing the EAPC of the CVD burden caused by HSI. *p* < 0.05 was considered significant. The R program (version 4.0.2) was utilized to perform all the statistical analyses.

## 3. Results

### 3.1. Deaths and ASMR of CVD Attributable to HSI

Globally, the number of CVD deaths attributable to high sodium intake increased by 41.08% over the past thirty years, from 1,215,851 in 1990 to 1,715,381 in 2019. However, the ASMR decreased from 33.16 (95% CI: 10.39, 69.96) to 21.51 (95% CI: 5.58, 47.2) per 100,000 population at the same time, with an average annual decline of 1.53% (95% CI: −1.57, −1.50).

In China, the estimated number of CVD deaths attributable to HSI in 2019 was 788,587.8, increasing by 53.91% compared with 512,346.4 in 1990. Meanwhile, the ASMR decreased from 68.52 (95% CI: 27.97, 119.98) to 42.46 (95% CI: 15.4, 76.86) per 100,000 population, and the annual trends of changes in ASMR were slightly slower than the global level, with an EAPC of −1.45 (95% CI: −1.55, −1.35). Notably, there was considerable gender inequality in the CVD burden due to HSI. The sheer numbers of CVD deaths in males were 1.54 times higher in 1990 and 1.98 times higher in 2019 than those in females, respectively. The ASMR between 1990 and 2019 was lessened from 52.16 (95% CI: 17.36, 100.5) to 26.86 (95% CI: 6.52, 55.72) per 100,000 population for females and reduced from 89.96 (95% CI: 39.84, 150.1) to 62.57 (95% CI: 24.39, 109.97) per 100,000 population for males. The percentage change in ASMR displayed a coincident downward trend in both genders, with EAPC being more pronounced in females than in males.

At the SDI region level, those with high-middle SDI experienced the highest number of CVD deaths attributable to HSI in both 1990 and 2019, while regions with high SDI had the complete opposite. At the same point in the timeline, after adjusting for the effects of differences in age structure, the ASMR was largest in regions with low-to-middle SDI and lowest in those with high SDI. EACP in ASMR differed by SDI quintile but presented a downward trend in all SDI regions in unison between 1990 and 2019, and it decreased the most in the high SDI regions and the least in the low SDI regions (Table 1).

At the provincial level, the CVD deaths attributable to HSI mainly occurred in Shandong, Henan, Hebei, and Sichuan, sharing more than 30% of the deaths in China in both 1990 and 2019. The fractional contribution of total CVD deaths caused by his ranged from 0.09% to 0.23% in 2019. Specifically, Tibet, Xinjiang, and Zhejiang were among the top three, while Gansu, Guangxi, and Hainan were ranked in the bottom three in terms of PAFs of HSI-related CVD deaths. The highest ASMR of CVD attributable to HSI was observed in Tibet at 105.86 (95% CI: 73.14, 265.75), Xinjiang at 82.32 (95% CI: 35.03, 142.37), and Qinghai at 71.14 (95% CI: 30.04, 124.28) per 100,000 population in 2019. In contrast, the lowest ASMR was found in Hong Kong at 13.9 (95% CI: 4.64, 27.92), Macau at 16.9 (95% CI: 5.64, 31.78), and Shanghai at 16.91 (95% CI: 5.31, 33.04) per 100,000 population in 2019. What is more, the heat map indicated that in 2019, the high values of ASMR were mostly concentrated in the provinces located in western and northern inland areas. On the contrary, the eastern and southern coastal regions of China exhibited generally lower ASMR values. During 1990 to 2019, the EAPC in ASMR of HSI-induced CVD decreased significantly in almost all provinces, and the most conspicuous decline was seen in Beijing (EAPC: −4.47, 95% UI: −4.78, −4.17), followed by Shanghai (EAPC: −3.85, 95% UI: −4.19, −3.5) and Jilin (EAPC: −3.23, 95% UI: −3.52, −2.94), while an upward tendency for ASMR was only observed in Gansu (EAPC: 0.56, 95% UI: 0.37, 0.75) and Xinjiang (EAPC: 0.38, 95% UI: 0.26, 0.5) (Appendix A; Figure 1A,C).

### 3.2. DALYs and ASDR of CVD Attributable to HSI

The global DALYs of high sodium-related CVD had an increase of 33.06% during the last three decades, from 304,684.79 in 1990 to 405,406.75 in 2019. Meanwhile, the ASDR decreased from 756.66 (95% CI: 262.0, 1495.6) per 100,000 person-years to 490.68 (95% CI: 146.3, 1010.7) per 100,000 person-years, with an average annual decline of 1.54% (95% CI: −1.57, −1.50).

In China, the estimated number of CVD DALYs attributable to HSI was 138,523.85 in 2019, marking a 39.39% rise from 193,086.28 in 1990. Still, the ASDR witnessed a decline from 1582.78 (95% CI: 751.7, 2585.9) per 100,000 person-years to 954.80 (95% CI: 425.7, 1601.0) per 100,000 person-years, and the annual trends of changes in ASDR were somewhat faster than the global level, with an EAPC of −1.61 (95% CI: −1.68, −1.53). Similar patterns in deaths and ASMR could also be observed in DALYs and ASDR in terms of gender. That is, males experienced a greater load of DALYs and ASDR and had a much slower decline in ASDR compared with females, with an EAPC of −1.13 (95% CI: −1.22, −1.03) in males versus an EAPC of −2.44 (95% CI: −2.53, −2.35) in females from 1990 to 2019.

At the SDI region level, the situation in DALYs, ASDR, and EAPC in ASDR was similar to the case in deaths, ASMR, and EAPC in ASMR of CVD caused by HSI we described before (Table 2).

### 3.3. Deaths and ASMR of CVD Attributable to HSI by Age and Gender

In 2019, the number of CVD deaths attributable to HSI peaked in groups aged 70–74 in both sexes, and it remained higher in males than in females until the age of 89, whereas the opposite was true for those aged 90 and above. The ASMR of CVD attributable to HSI approximated linear growth with age in groups younger than 90–94 years for both sexes (Ptrend < 0.05). In all age groups, the ASMR for males is higher than that for females, and the disparity between genders intensifies with age (Figure 2A). Between 1990 and 2019, the EAPC in ASMR showed a “V” relationship with age in each SDI region. Specifically, in low-middle and middle SDI regions, the trend of ASMR was downward in individuals aged below 74 and 84 and upward in individuals aged above 75 and 85, respectively, but the sharpest increase in those aged 85–89 and the steepest decrease in those aged 50–54 were commonly observed in both regions; in high-middle and high SDI regions, the EAPC in ASMR was below 0 across all age groups, and the most decline occurred in those aged 55–59 in both regions, with the latter regions having a larger decrease (Figure 3A).

### 3.4. DALYs and ASDR of CVD Attributable to HSI by Age and Gender

In 2019, the number of CVD DALYs attributable to HSI peaked in groups aged 65–69, regardless of gender, and it followed the pattern in deaths for males and females aged under and over 90. The ASDR of CVD attributable to HSI exhibited an elevated trend with age in groups younger than 90–94 years for both sexes and then decreased in males but continually increased in females (Ptrend < 0.05) (Figure 2B). At the SDI regional level, the temporal trend of ASDR in age groups resembled the pattern we described earlier for ASMR (Figure 3B).

### 3.5. The Association between ASMR and ASDR of CVD Attributable to HSI and SDI

In general, the ASMR of CVD attributable to HSI had non-linear associations with SDI from 1990 to 2019. The ASMR declined with the increment of SDI over time when the index was below 0.4 or above 0.7, while it tended to level off when SDI was between 0.4 and 0.7. The ASMR in the provinces of Xinjiang and Gansu steadily increased with the growth of SDI during the observation period. In spite of the overall downward trend of ASMR in Beijing, Tianjin, Guizhou, Hebei, Henan, Heilongjiang, Jilin, Liaoning, Shaanxi, Shandong, Inner Mongolia, Tibet, Xinjiang, and Qinghai, these regions still showed higher ASMR than expected based on comparisons of SDI for all years. Conversely, the ASMR in other provinces was below or at the same level as the expected values for all years. In addition, ASMR exceeded the expected values in examined the early stages of the time series for Hubei, Jiangxi, and Anhui, but it decreased as SDI increased. The relationship between ASDR and SDI shared the same patterns as that between ASMR and SDI (Figure 4).

A strong negative correlation was detected between the EAPC in ASRs of deaths and DALYs of and the SDI in 2019 (ρ = −0.659, *p <* 0.001 and ρ = −0.558, *p <* 0.001, respectively) (Figure 5C,D). However, no significant correlation was found between the EAPC in ASRs of deaths and DALYs and the ASMR or ASDR of CVD in 1990 (ρ = −0.123, *p* = 0.494 and ρ = −0.161, *p* = 0.370, respectively) (Figure 5A,B).

## 4. Discussion

Given the prevalent high-sodium diets and the observed patterns of exposure, this article exerts the first large-scale effort to conduct a comprehensive assessment of the epidemiological trends of CVD burden caused by HSI in China’s 33 provincial administrative units from 1990 to 2019, with a view to providing scientific evidence and thoughtful insights needed for efficiency in preventing the disease at a low cost.

In 2019, China presented a higher proportion of all CVD-related deaths and DALYs attributed to HSI compared to the global average. Salt has always maintained a difficult-to-shake presence in the dietary habits of Chinese residents, with nearly 70% of their intake of sodium originating from salt added during daily food preparation [17]. Despite a decline in salt intake from 16 g/d in 1991 to 10.4 g/d in 2015, credited to salt reduction efforts and health education campaigns, the issue of excessive-sodium diets remains problematic across China, considering the recommended upper limit given by the WHO [22]. China may need to make necessary adaptations based on current disease epidemiological trends associated with HSI exposure, seeking strategies that benefit the most widespread and vulnerable populations.

Our finding that, across all age groups, the CVD burden attributable to HSI was heavier in males than in females supports the observations in previous studies [23]. Males have higher intake and excretion of sodium compared to females, as reported in presumed studies, suggesting that levels of sodium exposure in the internal environment between males and females may well account for gender differences in HSI-related burdens [24,25,26]. Generally, men are more likely to engage in unhealthy lifestyles. A study conducted aboard deduced that males are more inclined to choose convenient dining options such as fast food and takeout [27]. In the domestic context of the evolving modern food system and modern marketing, males are also found to have a higher frequency of dining out for business and social purposes [28]. In fact, food eaten out is usually characterized by an unfavorable dietary profile. The sodium levels in Chinese restaurant dishes were revealed to be extremely high, with a single serving dish containing an average of 3331.2  ±  4156.9 mg sodium, which is almost 1.7 times the proposed intake for preventing non-communicable chronic diseases for Chinese adults [29]. In addition, social smoking and drinking, an entrenched part of Chinese culture, are also more prevalent among males, possibly as a result of social role differentiation. Cigarette and alcohol consumption, as identified risk factors, could also exacerbate the pathogenesis of CVD when combined with unhealthy diets. Another reason for lower HSI-related burdens in females is that they seem to be more efficient at maintaining Na^+^ homeostasis during acclimatization to dietary Na^+^ challenges, thereby reducing their vulnerability to the adverse effects of HSI exposure [30]. Premenopausal estrogen levels in females were also found to have a protective effect against CVD [31]. Therefore, gender-based sodium reduction policies in this regard should be urgently needed.

Although the ASMR and ASDR suggested a softened trend of CVD attributable to HSI from 1990 to 2019, the absolute number of its deaths and DALYs has increased substantially throughout the years in China. The growth and accelerated aging of the population may partly explain the findings. The ratio of China’s elderly, aged 60 and above, was reported to continually expand in the context of the country’s population growing from 1.1 billion to 1.4 billion, accounting for approximately 11.5% of the total in 2019, with expectations of reaching 16.9% by 2030 [18,32]. The aging population could act as a pivotal factor in the overall burden of diseases, including CVD caused by high-salt diets. We further explored that there was a greater burden of HSI-related CVD in the elderly compared to the youngsters, the explanations of which may be boiled down to several factors. Weakened blood vessel elasticity and heart function with aging could potentially predispose the elderly to the negative effects of HSI. Additionally, the implemented sodium reduction measures in China’s select counties and provinces may produce more pronounced effects on younger individuals. As for older adults, who have a consistent diet and less food consumption over a long period of time, these interventions could not significantly reduce their cumulative burden [33]. When the mitigation of the HSI-induced CVD burden was insufficient to counteract the impact of the worsening graying, a net increase in the burden would not be surprising.

Variations in the trends of ASMR and ASDR during 1990 to 2019 at the provincial level have been observed, showing complex associations with factors regarding nature and social setting. In 2019, Tibet was hardest hit by the HSI-induced CVD burden. The Tibet autonomous region is located on the Qinghai-Tibet Plateau, the highest plateau in the world with an average altitude of more than 4500 m, where most of the Tibetans are concentrated. A previous study among Tibetans exhibited a 2% increase in the prevalence of hypertension for every 100 m elevation, indicating such an environment is inherently to the detriment of subsequent cardiovascular outcomes [34]. Based on the unique climatic and geographical conditions there, the local inhabitants have dietary patterns peculiar to their group that feature a high proportion of meat, dairy products, and grain-based foods but a low proportion of leafy green vegetables and fruits, usually involving high levels of sodium but low levels of potassium [35]. This heightened sodium-to-potassium ratio is particularly concerning given the evidence that the synergistic effect of diets high in Na^+^ and low in K^+^ is greater than that of each per se on the pathogenesis of hypertension or CVD [36]. The lowest burden was found in Hong Kong and Macau. In fact, these places are also noted for people’s longevity. In addition to a sound healthcare system and strong public health measures, the populace is relatively well-educated, enabling them to have a full awareness of CVD and its risk factors and make informed behavioral changes [37]. This can be mirrored in their health-oriented diet to some extent. Residents there prefer to cook the ingredients in a more simplified way, like stewing or steaming, minimizing excessive processing and nutrient loss. Fresh fish, vegetables, and fruits, as essentials of local dietary preferences, have shown significant beneficial effects in reducing CVD risks in Hong Kong Chinese samples [38]. The result that the burden of CVD attributable to HSI was strongly contrasting between northern and Southern China confirmed the same finding in previous studies [39]. In terms of lifestyle issues, levels of sodium intake appear to play a dominant role in shaping the north–south differential for disease, with residents’ average daily sodium intake in Northern China being 2.32 g higher than that in Southern China [40]. In terms of environmental factors, ambient temperature and air quality could also influence CVD. While both high and low extremes increase mortality, the low-temperature effects on the disease persist for a longer duration [41]. The temperature difference between the two sides of the north–south boundary can be much greater in winter than in summer, fostering the contrasted disease burden. Additionally, due to extensive coal consumption driven by heavy industry development or winter indoor heating, the northern regions of China experience higher concentrations of PM 2.5 particles [42]. Interactions between PM 2.5 and HSI may co-exist, whereby prolonged exposure to PM 2.5 exacerbates the effect of HSI on CVD.

Wide disparities existed in HSI-caused CVD burden among regions with different SDI quintiles, reflecting disequilibrium in sodium intake control and medical healthcare. In our study, the ASRs of deaths and DALYs were comparatively the lowest in high SDI regions from 1990 to 2019. That makes sense, as residents in these regions have greater chances to avail themselves of quality education, well-established medical systems, and policy priorities, all of which are conducive to easing the disease burden. Interestingly, the correlation between the CVD burden and SDI was not monotonic or linear. In particular, the downturn in declines in ASRs of deaths and DALYs in low-middle SDI regions was more striking than that in middle SDI and high-middle SDI regions. This indicates that low-income regions, being the regions saddled with the heaviest health burden, are more likely to make a real difference by investing in diet enhancement. From 1990 to 2019, the annual decreasing trends in ASMR and ASDR were observed across all age groups for high-middle and high SDI regions, while the increasing trends were found at the end of the scale for low-middle and middle SDI regions. However, EAPC in ASMR or ASDR in different SDI regions uniformly showed a linear positive correlation with age ≥60 years, which may be due in part to the prolonged life expectancy by virtue of the reduction in deaths from common infectious diseases, leaving the number of elderly individuals much higher. Concerted efforts are required to address the outstanding CVD burden across China, especially in regions with low SDI and high sodium diets.

Aggregated information provided by representative datasets instead of the details of the total number and geographical distribution of those individuals allowed us to understand the broader epidemiological patterns and their overall impact of HSI on cardiovascular health outcomes across China. The interpretation of our study is still limited in several ways. Firstly, the analysis was reliant on the quantity and quality of the acquired data. Even though a good-quality vital registration system in China has provided adequate data on CVD mortality and morbidity in provinces, data for non-fatal outcomes of CVD may still be underreported for remote and poorer areas. Secondly, the profile of CVD burden in the urban–rural stratification has not been evaluated and examined due to data gaps. Thirdly, only individuals aged 25 and older were included in this study, while the details for younger groups are unavailable. Fourthly, the sodium-to-potassium ratio, which is detrimental for CVD onset, might potentially influence our findings, given its decrease from 4.1 to 3.1 between 1991 and 2015 in China [43]. Nevertheless, this decline was predominantly driven by a reduction in sodium intake, while potassium intake exhibited relative stability. Consequently, its effect on our results is expected to be slight. Fifthly, this study adopted a global perspective to analyze the CVD burden caused by HSI in China and underscored the benefits of nationwide salt reduction. Ecological studies may not establish causality and provide guidance at the individual level. Lastly, the statistical significance of the relationship between HSI and CVD might be affected by potential confounders, such as medication use and accessibility to medical resources, and interactions among dietary components that could not be adjusted promptly on account of data availability.

## 5. Conclusions

In summary, this study is grounded in a sizable and varied population hailing from different regions, ensuring sufficient statistical power for the analysis. This study demonstrated the profile of the CVD burden attributable to a high-sodium diet among Chinese residents between 1990 and 2019. Although a pronounced reduction in HSI-related CVD burden was achieved, the increasing number of deaths and DALYs still makes it an incremental public health issue over the past three decades. Males and the elderly bore a heavier disease burden in the context of the demographic transition. Diet management holds enormous potential to enhance public health, particularly in low-income regions. These findings call for prioritized public health interventions that are evidence-based and data-driven.

## Figures and Tables

**Figure 1 nutrients-16-01307-f001:**
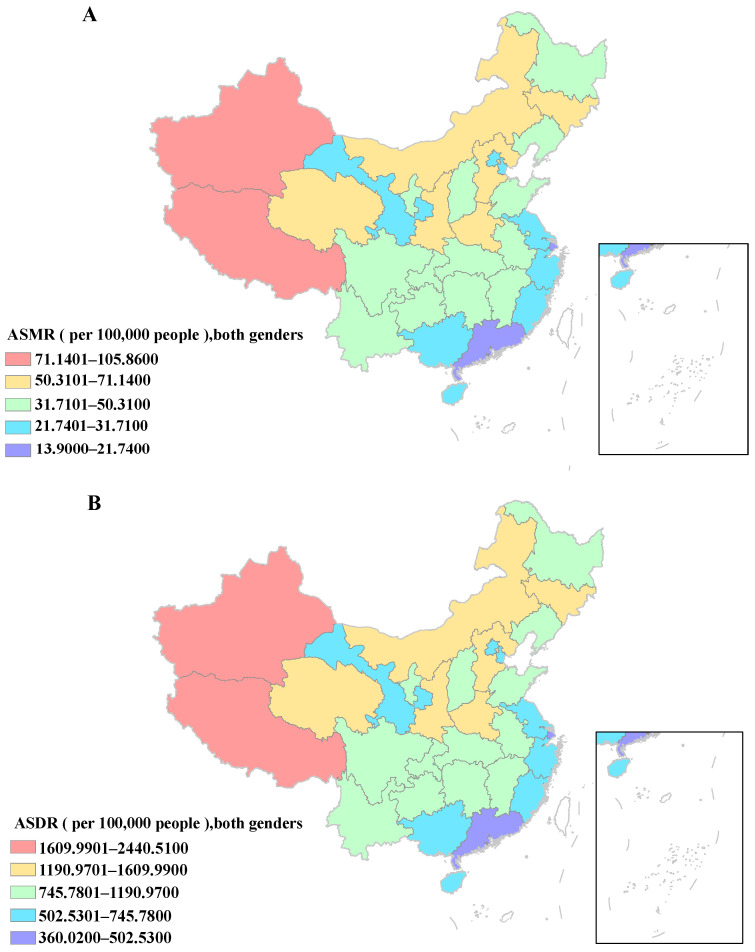
The burden of cardiovascular disease due to high sodium intake for both genders. (**A**) The spatial distribution of ASMR in 2019. (**B**) The spatial distribution of ASDR in 2019. (**C**) The EAPC in ASMR from 1990 to 2019. (**D**) The EAPC in ASDR from 1990 to 2019. ASMR, age-standardized mortality rate; DALYs, disability-adjusted life years; ASDR, age-standardized DALYs rate; EAPC, estimated annual percentage change.

**Figure 2 nutrients-16-01307-f002:**
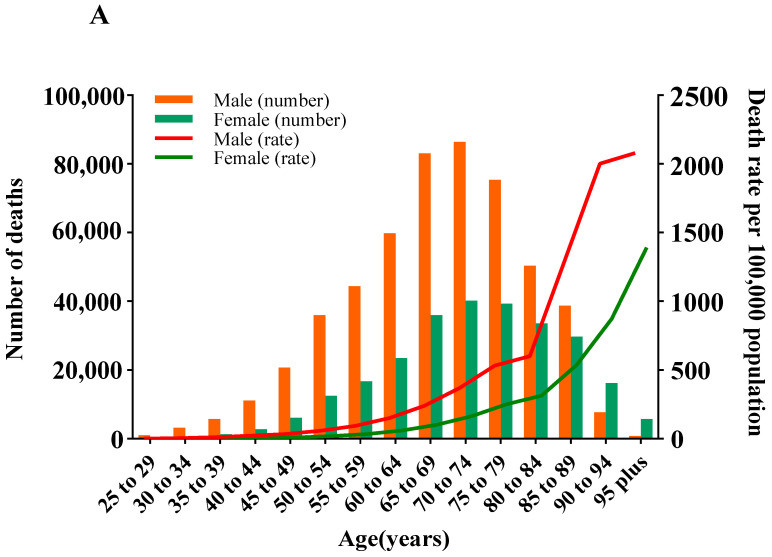
Numbers and rates of deaths and DALYs of cardiovascular disease due to high sodium intake across age groups by gender in 2019. (**A**) Deaths. (**B**) DALYs. DALYs are disability-adjusted life years.

**Figure 3 nutrients-16-01307-f003:**
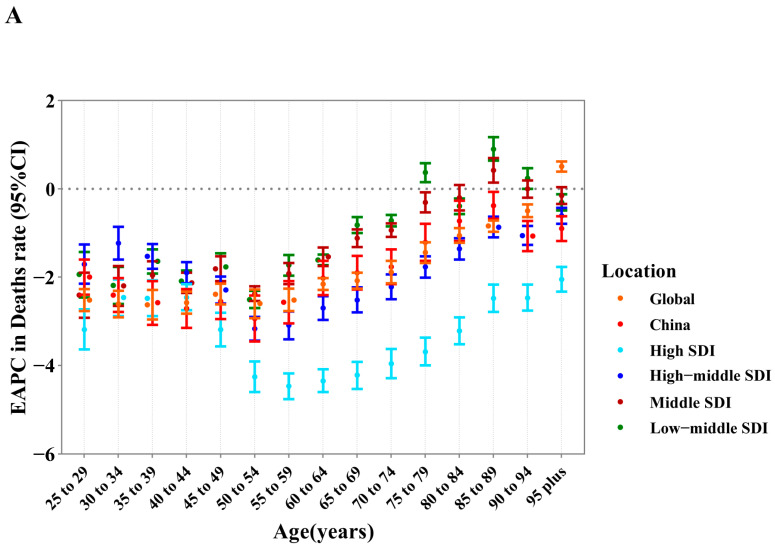
The temporal trends in burden of cardiovascular disease due to high sodium intake across age groups by location, 1990–2019. (**A**) EAPC in mortality rate. (**B**) EAPC in DALYs rate. EAPC, estimated annual percentage change. DALYs are disability-adjusted life years.

**Figure 4 nutrients-16-01307-f004:**
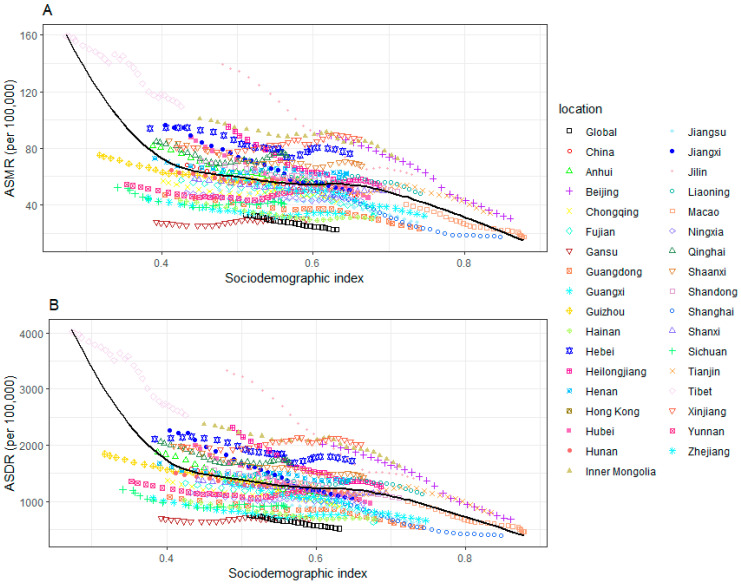
The burden of cardiovascular disease due to high sodium intake across 33 provinces by SDI for both genders, 1990–2019. (**A**) The correlation between ASMR and SDI. (**B**) The correlation between ASDR and SDI. SDI, sociodemographic index; ASMR, age-standardized mortality rate; ASDR, age-standardized DALY rate.

**Figure 5 nutrients-16-01307-f005:**
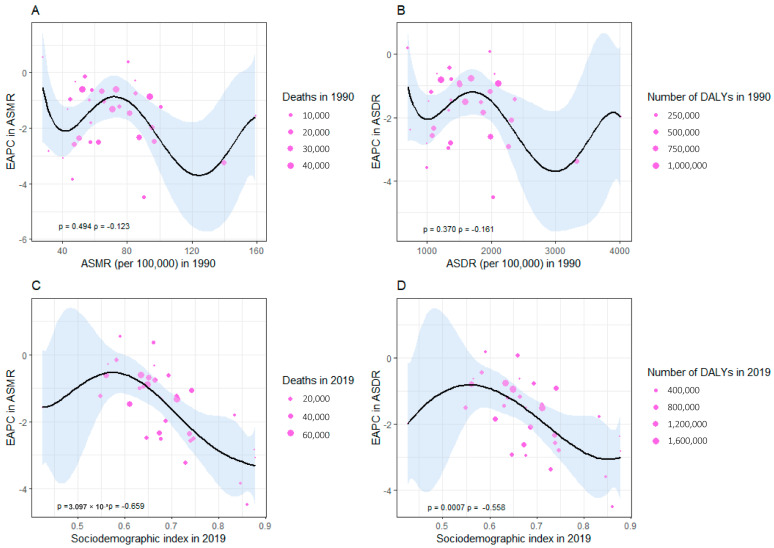
Factors influencing the EAPC. (**A**) The association between EAPC and ASMR in 1990. (**B**) The association between EAPC in ASDR and ASDR in 1990. (**C**) The association between EAPC in ASMR and SDI in 2019. (**D**) The association between EAPC in ASDR and SDI in 2019. Provinces were depicted by circles, with the size of each increasing according to the number of cardiovascular disease-related deaths and DALYs due to high sodium intake. EAPC, estimated annual percentage change; ASMR, age-standardized mortality rate; ASDR, age-standardized DALYs rate; DALYs, disability-adjusted life years.

**Table 1 nutrients-16-01307-t001:** Deaths and ASMR of cardiovascular disease due to high sodium intake in 1990 and 2019 and their percentage changes from 1990 to 2019.

Characteristics	1990		2019		1990–2019	
	Deaths Cases, No. (95% UI)	ASMR per 100,000 No. (95% UI)	Deaths Cases, No. (95% UI)	ASMR per 100,000 No. (95% UI)	PAFs % (95% UI)	EAPC (%) in ASMR No. (95% CI)
Global	1,215,850.67 (397,663.19 to 2,505,379.89)	33.16 (10.39 to 69.96)	1,715,381.13 (454,452.62 to 3,713,240.37)	21.51 (5.58 to 47.20)	0.09 (0.02 to 0.19)	−1.53 (−1.57 to −1.50)
China	512,346.4 (233,687.7 to 853,907.61)	68.52 (27.97 to 119.98)	788,587.8 (307,537.43 to 1,385,711.49)	42.46 (15.40 to 76.86)	0.15 (0.06 to 0.28)	−1.45 (−1.55 to −1.35)
Gender (China)						
Male	311,061.16 (151,828.83 to 490,285.48)	89.96 (39.84 to 150.10)	524,270.43 (225,211.66 to 876,595.25)	62.57 (24.39 to 109.97)	0.17 (0.07 to 0.29)	−0.96 (−1.08 to −0.83)
Female	201,285.24 (73,041.74 to 371,881.92)	52.16 (17.36 to 100.50)	264,317.36 (67,441.92 to 539,511.81)	26.86 (6.52 to 55.72)	0.12 (0.03 to 0.24)	−2.18 (−2.29 to −2.07)
Socio-demographic Index (SDI)						
High SDI	3772.18 (370.52 to 4293.03)	52.22 (22.38 to 88.22)	4575.81 (1609.17 to 6936.32)	19.16 (6.52 to 36.28)	19.16 (0.05 to 0.28)	−3.56 (−3.71 to −3.02)
High-middle SDI	20,309.93 (9732.94 to 28,932.90)	76.44 (30.92 to 132.48)	31,137.83 (12,152.52 to 49,705.38)	42.89 (14.48 to 84.42)	0.15 (0.05 to 0.27)	−1.88 (−2.23 to −1.63)
Middle SDI	16,908.47 (7177.02 to 28,677.93)	67.49 (25.61 to 121.99)	26,349.53 (9721.99 to 54,485.84)	48.46 (17.31 to 91.24)	5.98 (0.05 to 0.27)	−0.93 (−1.22 to −0.66)
Low-middle SDI	6740.50 (2819.47 to 7834.64)	87.39 (38.45 to 154.12)	10,841.83 (3600.87 to 20,705.91)	61.33 (23.33 to 111.58)	0.16 (0.06 to 0.28)	−0.74 (−0.88 to −0.60)

ASMR = age-standardized mortality rate; PAF = population attributable fraction; EAPC = estimated annual percentage change.

**Table 2 nutrients-16-01307-t002:** DALYs and ASDR of cardiovascular disease due to high sodium intake in 1990 and 2019 and their percentage changes from 1990 to 2019.

Characteristics	1990		2019		1990–2019	
	DALYs, No. (95% UI)	ASDR per 100,000 No. (95% UI)	DALYs, No. (95% UI)	ASDR per 100,000 No. (95% UI)	PAFs % (95% UI)	EAPC (%) in ASDR No. (95% CI)
Global	30,468,479.33 (10,829,442.92 to 59,419,298)	756.66 (262.03 to 1495.60)	40,540,674.51 (12,262,912.95 to 83,022,361.27)	490.68 (146.27 to 1010.73)	0.10 (0.03 to 0.21)	−1.54 (−1.57 to −1.50)
China	13,852,385.09 (6,916,321.04 to 22,011,605.74)	1582.78 (751.65 to 2585.91)	19,308,628.39 (8,894,786.58 to 31,852,222.15)	954.80 (425.86 to 1600.97)	0.19 (0.09 to 0.32)	−1.61 (−1.68 to −1.53)
Gender						
Male	8,608,041.4 (4,574,858.95 to 13,276,567.48)	2008.91 (1004.58 to 3173.03)	13,190,360.85 (6,551,116.67 to 21,235,740.84)	1358.90 (646.75 cto 2217.05)	0.22 (0.11 to 0.34)	−1.13 (−1.22 to −1.03)
Female	5,244,343.69 (2,247,167.81 to 9,012,808.73)	1196.65 (482.56 to 2093.97)	6,118,267.54 (1,979,661.12 to 11,147,594.56)	587.74 (185.00 to 1082.24)	0.15 (0.05 to 0.28)	−2.44 (−2.53 to −2.35)
Socio-demographic Index (SDI)						
High SDI	96,492.83 (88,944.24 to 198,188.75)	1185.88 (415.09 to 1959.03)	109,499.06 (62,723.97 to 218,135.01)	456.96 (205.66 to 826.28)	0.19 (0.09 to 0.31)	−3.33 (−3.58 to −3.07)
High-middle SDI	542,598.62 (286,974.19 to 869,579.72)	1820.67 (876.75 to 3011.52)	770,508.40 (350,208.69 to 1,320,252.39)	971.97 (378.71 to 1660.43)	0.19 (0.08 to 0.32)	−2.02 (−2.28 to −1.84)
Middle SDI	461,382.41 (220,529.43 to 851,330.99)	1591.61 (778.84 to 2691.05)	638,164.44 (290,164.88 to 989,275.51)	1074.08 (437.44 to 1876.64)	0.19 (0.09 to 0.32)	−1.20 (−1.41 to −0.95)
Low-middle SDI	192,364.27 (118,846.97 to 325,071.57)	2185.11 (1025.61 to 3633.15)	274,191.05 (106,753.79 to 491,301.09)	1412.95 (638.08 to 2380.01)	0.20 (0.09 to 0.32)	−1.10 (−1.23 to −0.98)

ASMR = age-standardized mortality rate; PAF = population attributable fraction; EAPC = estimated annual percentage change. At the provincial level, it was still Shandong, Henan, Hebei, and Sichuan that had the majority of CVD DALYs attributable to HSI, accounting for over 30% of the DALYs in China in both 1990 and 2019. The proportion of total CVD DALYs resulting from HSI ranged from 0.12% to 0.27% in 2019. Tibet, Xinjiang, and Zhejiang maintained their positions as the top three, while Gansu, Guangxi, and Hainan remained in the bottom three for PAFs. The highest ASDR of CVD attributable to HSI was observed in Tibet at 2440.51 (95% CI: 1274.33, 3772.27), Xinjiang at 1877.72 (95% CI: 934.52, 3041.33), and Hebei at 1609.99 (95% CI: 721.61, 2644.41) per 100,000 person-years in 2019. The lowest ASDR was detected in Hong Kong at 360.02 (95% CI: 149.85, 648.54), Shanghai at 381.45 (95% CI: 159.6, 662.46), and Macau at 442.08 (95% CI: 186.17, 730.45) per 100,000 person-years in 2019. Additionally, the heat map showed that China’s regional distribution variations in ASDR of CVD due to HSI were basically consistent with those in ASMR. What is also similar to the findings in ASMR is that almost all provinces had significant drops in ASDR of CVD attributable to HSI. Between 1990 and 2019, the top three still were Beijing (EAPC: −4.51, 95% UI: −4.81, −4.21), Shanghai (EAPC: −3.59, 95% UI: −3.89, −3.29), and Jilin (EAPC: −3.39, 95% UI: −3.65, −3.12), while ASDR in Gansu (EAPC: 0.18, 95% UI: 0.02, 0.34) continuously rose and remained stable in Xinjiang (EAPC: 0.07, 95% UI: −0.08, 0.23) (Appendix A; and Figure 1B,D).

## Data Availability

The authors declare that the data and material of this study are available for reasonable request. The data are not publicly available due to information restriction.

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
