# Peer review of "Cardiovascular Disease Burden Attributable to High Sodium Intake in China: A Longitudinal Study from 1990 to 2019"

_nutrients, 2024, doi:10.3390/nu16091307_

Round 1
Reviewer 1 Report
Comments and Suggestions for Authors
The aim of the paper by Liying Jiang et al. was to estimate the trends in high sodium intake-induced cardiovascular disease burden across China from 1990 to 2019. The authors concluded that the high sodium-induced cardiovascular disease burden is gender, age and socioeconomic-dependent.
The paper and the topic are interesting, and the conclusions seems appropriate, however several points must be clarified:
- First of all, the many acronyms make it difficult to read and interpret the paper. Tha authors need to reduce the number of acronyms.
- In the methods section the authors reported:” An average 24-h urinary sodium excretion”. What do they mean by average? Furthermore, the authors should provide more information on the 24-hour urinary collections and methods adopted for checking their completeness.
- The authors talk about cardiovascular mortality and morbidity but there is no mention of the cardiovascular risk factors present in the population studied, factors that should be taken into account in the analysis of mortality and morbidity
Comments on the Quality of English Languagemust be revised.
Reviewer 2 Report
Comments and Suggestions for Authors
This is a manuscript that describes a large-scale effort to highlight the association of high sodium intake with CVD across China's 33 provinces. The study is well-designed and supported by rationale provided in the Introduction section. Please find below the comments that in my opinion would strengthen the manuscript:
1. You mention that HSI was determined based on a 24-hour urinary excretion study. Perhaps providing more details regarding these collection procedures would be helpful (i.e. was it standard across all of China, and between 1990 and 2019?).
2. What is the total number of individuals included in this study? Also, what is this distribution across the 33 provinces?
3. What was the strategy to minimize confounding effects? Is it embedded in the analytical tool? A brief description of such would be helpful.
4. It has been suggested (i.e. by the WHO) that it is not so much the sodium but the sodium:potassium ratio that is detrimental for CVD onset. Na:K > 1 was associated with higher risk whereas, Na:K = 1 or <1 was protective. Where does the Chinese diet stand when it comes to potassium. I realize there may be not a clear cut answer to this question, but perhaps a comment on how does potassium in the diet get minded and whether it would be feasible to increase dietary K.
Reviewer 3 Report
Comments and Suggestions for Authors
The study conducted a nationwide analysis and visualized their results meaningfully. The findings are valuable for scientific insights and policy making. However, some questions should be answered before it is considered to be published.
The study utilized the GBD PAF equation to estimate the burden of HSI-related CVD. To enhance transparency and acknowledgment, the authors should provide the specific values of RR and the proportion of the population exposed to high sodium intake for both the global and Chinese contexts. This information would allow readers to understand the basis for their estimates better.
The noticeable difference in PAF between the global and Chinese populations, as evident in Table 1 and Table 2, warrants discussion. The authors should delve into potential reasons for the higher burden of HSI-related CVD in China. Factors such as dietary habits, lifestyle, and genetic predisposition could be explored.
The PAFs were not proportionate across the four SDI levels. The authors should elaborate on the reasons behind this disparity.
The authors should correct the labeling of Figures 3A and 2B in the main text, as indicated by the line numbers you provided.
While valuable for exploring associations, the study's ecological design inherently has limitations. These limitations should be explicitly addressed. For instance, ecological studies cannot establish causality at the individual level.
I raise an essential point regarding the accessibility to medical facilities as a potential confounder. Indeed, disparities in healthcare access can significantly impact CVD outcomes. The authors should acknowledge this limitation and consider its implications when discussing the relationship between HSI and CVD.
Round 2
Reviewer 2 Report
Comments and Suggestions for Authors
Fantastic work, very well done.
Author Response
Thank you for your thorough review and valuable feedback.
Reviewer 3 Report
Comments and Suggestions for Authors
The authors have responded to most of my comments thoughtfully. However, they addressed that data ownership constraints them from publicly sharing specific values from the original raw data, such as RR values and the proportion of the population exposed to HSI. I do not consider revealing these values to involve concern about the original raw data. Conversely, disclosing the analysis process is essential to science and ethics.
Author Response
Thank you for your thorough review and valuable suggestion again. I totally agree with you that disclosing the analysis process is essential to science and ethics. Also, it is true that this information of RR would allow readers to fully understand the basis for our estimates for HSI-related CVD burden. In fact, the GBD of China is one of the most important parts for the GBD study worldwide. That is, the collection of raw data was done by the group in China, while the Washington University was in charge of the calculation of our original raw data. Therefore, the details of RR values and the proportion of the population exposed to HSI were unavailable for us even if sharing specific values could help readers to consider the design and analysis process and to further underpin the estimation of the HSI-related CVD burden. We once again express our gratitude for your valuable comments. And, we ensure that all interpretations were based on these underlying data.